# Adipose Tissue Fibrosis: Mechanisms, Models, and Importance

**DOI:** 10.3390/ijms21176030

**Published:** 2020-08-21

**Authors:** Megan K. DeBari, Rosalyn D. Abbott

**Affiliations:** 1Department of Materials Science and Engineering, Carnegie Mellon University, Pittsburgh, PA 15213, USA; mdebari@andrew.cmu.edu; 2Department of Biomedical Engineering, Carnegie Mellon University, Pittsburgh, PA 15213, USA

**Keywords:** adipose tissue, fibrosis, in vitro models, in vivo models, biomaterials

## Abstract

Increases in adipocyte volume and tissue mass due to obesity can result in inflammation, further dysregulation in adipose tissue function, and eventually adipose tissue fibrosis. Like other fibrotic diseases, adipose tissue fibrosis is the accumulation and increased production of extracellular matrix (ECM) proteins. Adipose tissue fibrosis has been linked to decreased insulin sensitivity, poor bariatric surgery outcomes, and difficulty in weight loss. With the rising rates of obesity, it is important to create accurate models for adipose tissue fibrosis to gain mechanistic insights and develop targeted treatments. This article discusses recent research in modeling adipose tissue fibrosis using in vivo and in vitro (2D and 3D) methods with considerations for biomaterial selections. Additionally, this article outlines the importance of adipose tissue in treating other fibrotic diseases and methods used to detect and characterize adipose tissue fibrosis.

## 1. Introduction

Fibrosis is characteristically defined as the thickening and scarring of tissues from a pathological repair process. While often fibrosis is thought to occur following an injury, in adipose tissue changes in metabolism can trigger an inflammatory cascade that initiates the maladaptive fibrotic repair process. Once initiated, fibrotic changes in adipose tissue have devastating effects for patients. In most cases, adipose tissue fibrosis is undetected, difficult to reverse, and interferes with treatment options for obesity and other comorbidities.

White adipose tissue is responsible for storing and releasing energy in the body by regulating lipogenesis and lipolysis, respectively [1]. Lipogenesis is the process where free fatty acids and glycerol are taken up from the blood stream and stored in adipocytes as triglycerides. Conversely, lipolysis is the process in which triglycerides are catabolized into free fatty acids and glycerol that are released into the blood stream where they are used as an energy source by other organs [2]. An imbalance between these two processes leads to obesity and metabolic diseases, such as type 2 diabetes [3].

Obesity is associated with adipocyte enlargement (hypertrophy) and formation of new adipocytes (hyperplasia) [4]. This increase in adipocyte volume and tissue mass triggers inflammation, further dysregulation in adipose tissue, and eventually adipose tissue fibrosis [5]. Like other fibrotic diseases, adipose tissue fibrosis is the accumulation and increased production of extracellular matrix (ECM) proteins [6,7]. In healthy adipose tissue, the ECM can be remodeled to accommodate normal fluctuations in adipocyte size. However, when adipose tissue becomes fibrotic the stiff ECM cannot be dynamically remodeled.

Historically, obesity has been linked to an overall increase in lipolysis that results in ectopic accumulation of lipids in other tissues and insulin resistance [8,9]. An increase is observed when lipolysis is normalized to overall lean body mass; however, when normalized to lipid content there is a decrease in basal lipolysis rate per cell [10,11,12]. Furthermore, starved hypertrophic adipocytes only lose a portion of their stored triglycerides (<50%) without supplemental factors [13]. There is limited literature investigating the effect adipose tissue fibrosis has on lipolysis, so most studies assume the same trends as obesity. There are several theories as to why this may not be valid. One being that as adipocytes shrink, stress between the ECM and cells can increase, eventually inhibiting lipolysis, leading to enlarged adipocytes without a mechanism for depleting their lipid stores [7].

With decreased lipolytic function, it is not surprising that fibrotic adipose tissue has been linked to difficulties losing fat mass [14] and insulin resistance [15]. For example, gastric bypass surgery, which generally causes rapid fat loss and increases in insulin sensitivity, is less effective for patients with high degrees of adipose tissue fibrosis [14,16]. Specifically, insulin resistance is linked to higher levels of adipose tissue inflammation and increased ECM deposition [15]. Initially, as insulin sensitivity decreases, adipocytes begin to undergo enhanced lipolysis and release free fatty acids into the environment. These free fatty acids cause further insulin resistance and can lead to inflammation by signaling macrophages and other immune cells to the area [17]. Over time this inflammation increases ECM protein production leading to interstitial fibrosis in adipose tissue. The increased ECM stiffness prevents the adipocytes from expanding in a healthy manner, causing the tissue to be metabolically dysfunctional, including adipocyte death, decreased lipolysis, and disrupted cell-cell interactions [6,14].

Adipose tissue fibrosis has a complicated and critical role in metabolic dysfunction and obesity. With the growing obesity epidemic, it is important to understand the mechanisms behind the development and progression of adipose tissue fibrosis. Furthermore, targeted therapeutic treatments are required for patients with fibrotic adipose tissue. This article discusses recent research in modeling adipose tissue fibrosis using in vitro (2D and 3D) and in vivo methods with considerations for biomaterial selections. Additionally, this article will outline the importance of adipose tissue in treating other fibrotic diseases and methods used to detect and characterize adipose tissue fibrosis.

## 2. Mechanisms

While there are a number of cell types, environmental factors, and cellular pathways that contribute to the development of adipose tissue fibrosis, the overarching cause is obesity (Figure 1) [18,19,20,21,22,23]. However, the link between obesity and fibrosis is complex and incompletely understood. While there is an increased prevalence of adipose tissue fibrosis in cases of severe obesity [19,24], not all obese patients will develop fibrotic tissue. Primarily based on the correlative nature of human studies, there remains many open questions surrounding the link between obesity and fibrotic tissue. This synopsis aims to introduce several mechanisms that have been thoroughly researched.

Proper adipose tissue ECM composition and remodeling is crucial for cellular function. During the development of fibrosis, excess ECM proteins are produced while ECM degradation is limited. Several cell types in adipose tissue including adipocyte progenitors, adipocytes, fibroblasts, and myofibroblasts are responsible for the production of ECM proteins [25].

Fibronectin and collagens are the most prevalent ECM proteins found in healthy and fibrotic adipose tissue. Comparing obese and diabetic mice, it was determined that type I, III, V, and VI collagens are present in higher concentrations compared to healthy mice [26]. Collagen I (COL1) is responsible for providing structure within the tissue [14], while collagen V (COL5) in high concentrations inhibits angiogenesis in adipose tissues [27]. Specific to adipose tissue fibrosis, collagen VI (COL6) has been investigated for its fibrotic role, with the absence of COL6 (in a knockout model) resulting in lower inflammation and uninhibited enlargement of individual adipocytes [28]. Furthermore, a component of COL6, COL6a3 named endotrophin, stimulates fibrotic collagen deposition, inflammation, and insulin resistance [22]. In times of energy surplus, adipocytes undergo hypertrophy (enlargements of cells) and hyperplasia (an increase in the number of cells) [29]. As adipocytes become larger, they not only exhibit morphological changes but also become metabolically dysfunctional [30,31], producing several proinflammatory chemokines. These chemokines include tumor necrosis factor-alpha (TNFα), inducible nitric oxide synthase (iNOS), interleukin 6 (IL-6), interleukin 8 (IL-8), C-reactive protein, Transforming growth factor beta 1 (TGFβ1), soluble intercellular adhesion molecule (ICAM), and monocyte chemoattractant protein 1 (MCP-1) [31,32,33,34,35,36,37,38,39,40]. Additionally, adipocytes exhibit increased gene expression for ECM production under a high fat diet [41]. Similar to mature adipocytes, adipose derived stem cells (ASCs) also undergo significant changes during obesity. After being fed a high fat diet, mice had higher concentrations of ASCs that expressed platelet-derived growth factor receptor α (PDGFRα) [42]. PDGFRα activation opposes adipogenesis and causes cells to differentiate into a profibrotic phenotype [43]. The PDGFRα^+^ ASCs were found to promote fibrosis and are associated with insulin resistance [42].

Hypoxia is another major contributor in the development of adipose tissue fibrosis. As adipocytes expand due to lipid accumulation, they eventually reach the diffusion limit of oxygen. This results in hypoxia that causes stress signals to increase angiogenesis and remodel ECM proteins in an attempt to mitigate the oxygen shortage. In hypoxic adipose tissue, hypoxia-inducible factor 1-alpha (HIF1α) is expressed. Instead of initiating proangiogenic conditions in adipocytes, it enhances synthesis of ECM components. HIF1α is also suggested to play a role in collagen crosslinking and stabilization [7]. Both adipocytes and adipocyte progenitor cells produce angiogenic factors, such as vascular endothelial growth factor (VEGF), leptin, fibroblast growth factor 2 (FGF-2), and hepatocyte growth factor (HGF) [44,45]. Interestingly, endothelial cells derived from obese adipose tissue have increased gene expression related to inflammation and senescence, and impaired angiogenesis [46], suggesting that initial signals to promote angiogenesis are reversed if hypoxia is unresolved. Therefore, prolonged hypoxic conditions cause injury and long-term damage, fibrosis, cellular senescence, and necrotic adipocyte death [47].

Unresolved hypoxia can also attract proinflammatory cell types to the area. A population of cells that have a high concentration in fibrotic adipose tissue are M1 macrophages. Resident macrophages comprise between 10–15% of stromal cells in the adipose tissue of healthy, lean individuals [48]. This concentration jumps to between 45–60% of stromal cells in obese individuals [35,49,50]. Macrophages are typically observed in “crown-like structures” surrounding dead or dying adipocytes in obese adipose tissue [51,52]. The increase in macrophage concentration causes an increased recruitment of monocytes to the area; where the monocytes then differentiate into macrophages [35,48]. In the presence of M1 macrophages, ASCs are predisposed to differentiate into proinflammatory cells. This change occurs due to pro-inflammatory cytokines secreted by the macrophages [49,53], where it has been shown that they are the primary producers of TNFα, iNOS, and IL-6 [44].

ECM degradation is regulated by metalloproteinases (MMPs) and tissue inhibitors of metalloproteinases (TIMPs). MMPs have the ability to solubilize ECM components while specific TIMPs are able to inhibit MMPs activity. Obesity and type 2 diabetes have been linked to increased circulating TIMPs concentration causing MMP activity to be lower than in healthy individuals [54,55]. The increase in proinflammatory cytokines that occurs during hypoxia and inflammation has also been linked to regulation of MMPs expression [56].

Another cell population that has been linked to adipose tissue fibrosis is mast cells [57,58]. Mast cells can promote fibroblast growth and collagen production by releasing cytokines, chemokines, proteases, etc. [59,60]. By comparing adipose tissue from obese patients with the metabolic syndrome to healthy patients, it was determined that there is an accumulation of mast cells in the subcutaneous adipose tissue of obese patients. Related to fibrotic changes, there were positive correlations to collagen, leptin, and glucose concentrations, as well as waist circumference [57,61]. Overall, more research is needed to determine the role of mast cells and macrophages in the fibrotic transition of adipose tissues.

One method of better understanding the role of different cell types in ECM production, immune reactions, and fibrosis is single-cell RNA (scRNA) sequencing. When looking at adipocytes specifically single nuclei adipocyte RNA sequencing (SNAP-seq) is used [62]. These techniques have been used to better understand variability in cells from different fat deposits or pathological settings [63,64,65,66,67]. It offers a method of characterizing different subpopulations of adipose tissue cells and detecting heterogeneity in the populations [68].

## 3. In Vitro Adipose Tissue Fibrosis Modeling

Whenever possible, in vitro models are utilized to refine, reduce, and replace animal models. In vitro models can be quickly developed (compared to animal models), are easily manipulated with highly defined conditions, and have a high likelihood of clinical translation (when developed with human cells). Having accurate biomimetic in vitro adipose tissue fibrosis models can serve as useful tools for generating a better understanding of the mechanisms behind fibrosis development and progression, as well as drug screening for effective anti-fibrotic drugs.

### 3.1. 2D Models

White adipocytes have large unilocular lipid droplets in their cytoplasm. This causes the mature adipocytes to float in media when cultured using 2-dimensional (2D) approaches. Unable to meet their nutritional requirements, the floating adipocytes will lyse in a matter of days [69]. Techniques have been developed to combat this unique problem, including a method termed “ceiling culture” where flasks are filled with media and adipocytes attach to the ceiling of the cell culture flask [69,70,71]. However, cells cultured in this way do not function the same as in vivo adipocytes and lose their lipid stores [72]. As an alternative to culturing mature adipocytes ex vivo, ASCs are differentiated on 2D substrates to an immature multilocular phenotype (full differentiation cannot be achieved in these systems as lipid laden cells will detach from the surface).

Two-dimensional culture systems have been used to model cellular changes during obesity and adipose tissue fibrosis. Many systems use macrophages to mimic the inflammatory effects associated with obesity that trigger adipose tissue fibrosis [53,73,74,75,76,77]. One model system used media that was conditioned by macrophages from an obese human patient with or without supplementation of lipopolysaccharide (LPS), to study the secretory effects on ASCs from a healthy patient. The results indicated that ASCs have a lower capacity to differentiate in the presence of macrophage byproducts. Additionally, in the presence of LPS the ASCs became proinflammatory and secreted higher concentrations of inflammatory factors [77]. Following a similar experimental setup, another study found that ASCs exposed to factors secreted by M1 macrophages increased ECM remodeling. The ASCs had a proinflammatory phenotype, increased proliferation and migration, but a decreased ability to differentiate [53]. Together, these results indicate that the presence of secretory factors from M1 polarized macrophages primes ASCs to develop a proinflammatory phenotype that is pro-fibrotic, rather than pro-adipogenic.

Another study corroborated these results with a co-culture model. Proinflammatory macrophages (CD14+) and human ASCs (hASCs) were investigated to determine how stem cell differentiation was affected by their interaction. Consistently, there was a significant decrease in differentiation after 14 days in the presence of proinflammatory macrophages, with a decrease in adiponectin, CEBPβ, and GLUT4 and an increase in IL6 gene expression [74]. These results have an increased physiological relevance compared to similar studies [5,75,78] because both cell types were sourced from the same patient. It was proposed that the cytokine levels would be similar to native human adipose tissue because the same cell source was used. This suggests that 2D models could be used to study other fibrotic mechanisms related to ASC commitment and proinflammatory factors.

### 3.2. Three-Dimensional Models

#### Biomaterials Used for 3D Adipose Tissue Culture

Due to the decreased physiological relevance of culturing adipocytes using 2D techniques, 3D scaffolds are often used to grow and model adipose tissue. Several synthetic and natural biomaterials have been used to successfully culture ASCs and whole adipose tissue ranging from synthetic to natural polymers (Table 1). While some progress has been made modeling fibrotic adipose tissue using 3D in vitro platforms, this area of research is still at early stages of development with large potential for growth.

Diabetic patients have higher degrees of adipose tissue fibrosis [79]. To better understand the link between the ECM and adipocyte metabolic regulation, researchers used human decellularized adipose tissue from patients with and without type II diabetes mellitus as a model system. Combining ECM from a nondiabetic patient and adipocytes from a diabetic patient resulted in adipocytes that had fully restored basal lipolysis and insulin stimulated glucose uptake. Conversely, combining ECM from a diabetic patient and adipocytes from a nondiabetic patient decreased insulin sensitivity, but had no effect on lipolysis [80]. This suggests that dysfunctional fibrotic cues are found in the matrix and are not an intrinsic property of the cells.

The interaction of matrix stiffness and architecture with adipocytes was also tested in an in vitro model. Researchers incorporated ethylene glycol-bis-succinic acid N-hydroxysuccinimide ester (PEGDS) into a collagen hydrogel system. The addition of PEGDS created a more organized and interconnected ECM by establishing intrafibrillar and interfibrillar crosslinks. This was used to investigate the effect mechanical changes, like those that occur during adipose tissue fibrosis, have on adipocyte morphology and function. In stiffer hydrogels, adipocytes experienced increased profibrotic gene expression and ECM deposition. By inhibiting actin contractility, the researchers determined that the adipocyte dysfunction was regulated by actin cytoskeletal stress fibers that registered the mechanical properties of the surrounding environment [81]. This research indicates that if fibrosis could be reversed cells could be rescued from the fibrotic phenotype. Additionally, this research further supports that fibrotic cues are found in the extracellular matrix.

To mimic in vivo fibrosis in another system, researchers created hypoxic clusters of ASCs that grew into 3D cell masses comprised of ASCs, alpha smooth muscle actin positive (αSMA-positive) cells, necrotic cells, and collagen [82]. Researchers compared these 3D cell masses to 2D cultures and found that there was a significant increase in TGFβ1 secretion, hydroxyproline concentration, and lactate dehydrogenase secretion. These changes indicate an increase in expression of the fibrotic phenotype, ECM production, and cell death which align with physiological changes that occur during adipose tissue fibrosis development [82].

### 3.3. Biomaterial Considerations for In Vitro Modeling

When designing a biomaterial for modeling adipose tissue fibrosis, evaluating biomaterials that successfully grow healthy adipose tissue, as well as models that are used for other fibrotic tissues, can be a good starting point. Important biomaterial characteristics that should be considered follow (Figure 2).
*Stiffness*: Stiffness is an intrinsic property of a material and is defined as the resistance of an elastic material to deformation by an applied force. Adipocytes and ASCs are mechanosensitive and mechanoresponsive [100,101]. In the body, mechanical forces are balanced internally by the cytoskeleton and externally through the ECM. Scaffolding materials act as the ECM in in vitro models, therefore, the stiffness should be carefully considered as it will influence the cellular cytoskeleton and phenotype [100]. Scaffold stiffness can be controlled by adding other polymers or additives [81]. A recent study found that stiffer biomaterials triggered fibrotic traits in adipocytes, such as increased profibrotic gene expression and ECM deposition [81]. Additionally, stiffer matrices were found to promote osteogenic differentiation of ASCs, while softer matrices promoted adipogenic commitment [101].*Viscoelasticity:* Under deformation, a viscoelastic biomaterial will exhibit both elastic and viscous behavior making it a time-dependent response. Similar to stiffness, viscoelasticity is an intrinsic material property. The stress response of the material will vary based on strain and history of deformation and allows viscoelastic materials to exhibit stress relaxation, hysteresis, and creep. Soft tissues, like adipose tissue, are composed of solids and liquids and naturally behave like viscoelastic materials. An increase in collagen content results in a greater elastic component to the adipose tissue, and therefore fibrotic tissues would exhibit a decrease in the time-dependent viscous behavior.*Degradability:* A biomaterials degradability is also an intrinsic trait. It is dependent on the molecular properties of the biomaterial. Factors, such as cross-linking, can decrease the degradability. As fibrosis involves significant deposition of ECM proteins, degradability should be considered to ensure that cells have enough time and the ability to remodel their environment before the scaffolding degrades.*Dimensionality*: Dimensionality is an extrinsic material property that defines the number of dimensions an object occupies. To make the most accurate model for adipose tissue fibrosis a 3-dimensional model is necessary. Typically, growing cells in 3D constructs is more difficult than 2D, but due to adipocytes’ unique buoyancy issues, and the fact that 2D cultures results in different morphological (multilocular lipid droplets as opposed to the single unilocular lipid droplet observed in vivo) and functional adipocytes, 3D culture is often pursued as a more physiologically relevant option. In particular, the largest difference is in cell volume; 2D culture results in adipocytes with less than 20% the total cell volume of lean subcutaneous adipocytes and less than 3% the total cell volume of obese subcutaneous adipocytes [102,103,104,105,106].*Deformability*: Deformability is an intrinsic material property that is defined as the ability for the material to change shape. The goal of designing a biomaterial to model adipose tissue fibrosis is to encapsulate adipocytes and the stromal vascular fraction (SVF) in a scaffold with properties similar to the ECM found in adipose tissues. One method is to begin with a highly deformable biomaterial that has similar properties to healthy adipose tissue and allow cells to remodel their environment by triggering fibrosis. Alternatively, by beginning with a biomaterial scaffold that has properties of fibrotic adipose tissue, and a low degree of deformability, fibrotic adipose tissue can be modeled. Depending on the experiment and the end goal, each method offers unique advantages.*Plasticity*: Plasticity is an intrinsic material property that is related to stiffness and viscoelasticity. Several studies have found that cells are able to plastically remodel certain biomaterial environments, such as collagen and fibrin gels [107,108,109]. Cells align and compact the fibers around them to the point where, when the cells are removed, voids remain [110]. Healthy adipose tissue is a dynamic organ that is remodeled constantly to allow fluctuations in adipocyte size to meet energy storage and demand needs. However, adipose tissue fibrosis limits the capacity of adipose tissue to remodel [111]. The addition of crosslinks can lower the accessibility of cells to remodel the biomaterial [112].*Porosity*: Porosity is an extrinsic property that is defined as the percentage of void space in a material. Generally, biomaterial scaffolds used to culture adipose tissue have high porosities (>90%) [83,84] to accommodate the large size of lipid-laden adipocytes. However, pore sizes vary considerably based on the method of formation (135–633 µm) [83,93]. For example, gas foaming results in a larger range of pore sizes and is more difficult to control compared to salt leaching [113]. Researchers that have used salt leaching to generate their scaffolds have chosen pore sizes ranging from 500–600 µm [93]. Electrospun scaffolds had slightly lower porosity (~88%) and considerably smaller pore sizes (6–70 µm); however, these scaffolds were used with murine ASC cell lines that are smaller and do not contain the large unilocular lipid droplet [89]. Three-dimensional printing can be used to create specific pore sizes but depends heavily on the print resolution.*Processing:* There are many methods of processing biomaterials. Some researchers have used electrospinning as a method of making scaffolds for in vitro adipose tissue modeling [89,90]. Electrospinning was chosen because the resulting scaffold has similar structure to collagen fibers found in the ECM matrix of adipose tissue. However, electrospinning resulted in lower porosity and smaller pore sizes. Three-dimensional bioprinting has been used to model other fibrotic diseases [114]. Printing with cells allows for more complex tissues to be created. Currently, printing vasculature is difficult, which limits the size of the print, as necrotic cores develop if cells do not have access to nutrients in large tissue constructs without vasculature. Three-dimensional printing also is not an option for all biomaterials. Decellularized tissue matrices have been used to study adipose tissue fibrosis [80], with conduits for vasculature to help sustain long term culture. However, decellularized tissue matrices can limit the cell-cell interactions and only represent the final stages of the disease [115]. Hydrogels are useful for tissue culture due to ease of fabrication and can be made using ECM proteins [81]. However, the mechanical properties of hydrogels have lower tunability.*External Mechanical Forces*: Including mechanical forces, such as tensile or compressive strains, is independent of material properties. Applying tensile strains of about 12% to adipocyte cultures resulted in faster accumulation of lipids and larger lipid droplets compared to adipocytes cultured under no external stresses [116]. This could be a useful technique to expedite the formation of a lipid-rich model.

## 4. Current Methods of Modeling Fibrotic Adipose Tissue In Vivo

### Rodent

Animal models have been extensively used in obesity research. While it is beyond the scope of this review to delve into all animal models of obesity, many excellent reviews exist [117,118,119]. To investigate adipose tissues dynamics, mice are the most common animal model used. Mice are advantageous as they are readily available with well-established protocols for the development of monogenic (single genetic cause) and polygenic (surgical, chemical, dietary, or environmental causes) models [117], comparative historical data, a short lifespan and breeding cycle, and the ability to procure genetically identical strains [120]. Murine models can provide important whole-body information and have served as a useful tool in learning about adipose tissue fibrosis. Particularly relevant for exploring the dynamics of fibrotic adipose tissue, systemic effects (i.e., dynamic signaling with muscle, liver, etc.) and immune interactions can be explored that are not feasible in in vitro models.

Despite the extensive use of murine models, there are some significant drawbacks. For example, modeling obesity in mice requires dietary, genetic, and chemical modifications that limit their applicability to human translation [117]. Additionally, there are significant differences in location and purpose of fat deposits. Most murine research uses perigonadal fat pads as the adipose tissue source; however, humans do not have an analogous fat deposit [121]. Finally, differences between male and female mice and lipolysis mechanisms are not translatable to humans [121]. For example, adult female fat deposits are strongly influenced by reproductive hormones, while this is not seen in female rodents. Additionally, mice and human adipocytes respond differently to some lipolytic agents [122,123,124].

Among some of the most common animal models of obesity are: the leptin-deficient *ob/ob* mouse [125,126,127]; the leptin receptor deficient *db/db* mouse [125,128] and (the more specific) *ss/ss* mouse [129,130]; rat analogs with mutated leptin receptor domains: the Zucker and Koletsky rats [131,132]; deficits downstream from the leptin receptor: the proopiomelanocortin knockout [133,134], the melanocortin 4 receptor deficient mouse, and the Agouti related protein knockout [135]; and models of diet induced obesity [117]. Depending on the study design each model has advantages. Diet induced obesity in animals likely mimics human obesity more than genetically modified models (as mutations are very rare in humans) and are therefore a good fit for prospective therapeutics [117]. High fat diet models (>30% of energy from fat) are highly prevalent as dietary fat intake increases adiposity [136]. In particular, many diet-induced obesity models either use fat- and sugar-rich supermarket foods (cafeteria diet) or focus on saturated fatty acids and simple sugars (western diet). On the other hand, transgenic or spontaneous mutations are beneficial for exploring the role of specific molecular targets in the progression of obesity.

Obesity is closely linked to fibrosis, and thus many animal models of obesity naturally develop fibrosis. For example, leptin deficiency, as occurs in the *ob/ob* mouse results in adipose tissue fibrosis [28,137]. Likewise, collagens are highly upregulated in adipose tissue during metabolic challenges in the *db/db* mouse [28]. Currently, obese models of rats are being used to study fibrosis in other tissues, such as the pancreas (Zucker rats + high fat diet) [138] and liver (wistar rats + high fat diet) [139]; however, more research needs to focus on adipose tissue fibrosis in obese animal models.

Knockout models and overexpression models of genes are well-suited to study specific molecular targets involved in adipose tissue fibrosis. One study targeted periostin, an ECM protein that is secreted following a high fat diet in mice that amplifies inflammation, collagen cross-linking, and degradation of the ECM. In periostin knockout mice, there were fewer crown-like structures and reduced fibrosis compared to wildtype mice [140]. In another study, the same profibrotic genes (COL1A1, COL6A3, Lumican (LUM), Tensascin C (TNC)) overexpressed in humans with adipose tissue fibrosis [141] were overexpressed in C3H mice. In conjunction with a high fat diet, the C3H mice exhibited insulin resistance, collagen overproduction, increased macrophage activity, and adipocyte metabolic dysfunction. It was also determined that an important factor to consider in adipose tissue fibrosis models is the strain of mouse used, as different strains vary in levels of gene expression. Compared to the C57BL/6J mouse strain, which is often used in obesity research, the C3H strain showed earlier profibrotic gene expression [141], indicating it was a more accurate model of fibrosis. In another study, researchers studied gender differences with a knockout model. In mice, female adipose tissue has higher insulin sensitivities, less susceptibility to inflammation, and a higher expression of estrogen receptors, compared to male adipose tissue [142,143,144]. Additionally, in mice, adipose tissue estrogen receptor α (ERα) regulates body fat distribution, inflammation, and fibrosis [145]. They found that ERα knockout (αERKO) mice had enlarged adipocytes and higher degrees of inflammation and fibrosis in both males and females compared to wildtype [145]. Notably, female αERKO mice had a significantly higher expression of the COL6 gene, a gene related to ECM production and fibrosis. Taken together, these models highlight the important role of key proteins in the pro-fibrotic profile.

Another approach to model fibrosis is to target microRNAs (miRNAs), which are small noncoding RNAs that regulate gene expression [146,147]. Inflammatory and dysfunctional metabolic processes related to obesity have been linked to miRNAs. Specifically, it has been found that miRNA155 (miR155) plays an important role in adipocyte differentiation into the white phenotype and activation of proinflammatory pathways [148,149]. By deleting miR155 and feeding male mice a high fat diet to induce obesity, it was found that this deletion caused mice to store less visceral adipose tissue but exacerbated adipose tissue fibrosis compared to wildtype mice fed a high fat diet [150]. This was surprising because other groups have shown that miR155 plays a role in promoting macrophage polarization to the M1 phenotype [151,152,153]. By preventing the polarization of macrophages to the proinflammatory phenotype, it was thought that the prevalence of adipose tissue fibrosis would be decreased. It is important to note that the combination of miRNA155 deletion and a low-fat diet did not result in adipose tissue fibrosis. Conflicting results were gathered when a similar study was performed using female mice. Feeding miR155 female knockout mice a high fat diet resulted in lower degrees of obesity. Additionally, these mice had reduced inflammation and cell hypertrophy [154]. These results further illustrate that adipose tissue is sexually dimorphic.

The long-term goal of modeling fibrosis in animal models is not only to explore mechanisms of fibrosis but also to target treatments. Knockout models can be combined with traditional models of obesity to explore therapeutic targets. For example, knockout of collagen VI was explored in the *ob/ob* environment and with exposure to a high fat diet [28]. Interestingly, collagen VI deficiency resulted in an improvement of the metabolic phenotype in both the high-fat diet and *ob/ob* background. Similarly, in another knockout model, iNOS ablation in leptin-deficient mice (*ob/ob*) decreased fibrosis and metabolic dysfunction [137]. Towards the goal of clinical translation, a murine adipose tissue fibrosis model was used to investigate the effect of drugs on modulating adipose tissue fibrosis in a diet induced obesity model. In another study, Isoliquiritigenin (ILG) was used to treat adipose tissue fibrosis in a C57BL/6 diet induced obesity model. Mice fed a high fat diet and given ILG showed reduced fibrotic area, TNFα, COL1, and TGFβ1 expression compared to control mice. Importantly, there were no significant differences between high fat diet plus ILG and normal diet [155]. Overall, these studies emphasize that while rodent models may not directly translate to human clinical outcomes, they provide an important pre-clinical tool for controlling confounding variables and determining systemic interactions, immune responses, and genetic influences.

## 5. Human Epidemiological Studies

Human epidemiological studies are essential for correlating mechanistic data from in vitro and in vivo studies to fibrotic outcomes clinically. However, human studies on adipose tissue fibrosis are limited and inconsistent.

In human adipose tissue fibrosis studies, one of the most well studied comorbidities is the effect of diabetes. One study examined the effect of bariatric surgery on diabetic versus insulin-sensitive patients. Using patients with similar age, body mass index, and fat mass, the study showed that before surgery diabetic patients had a significantly higher degree of adipose tissue fibrosis. Six months after bariatric surgery, fibrosis levels had not decreased, even with significant weight loss [79]. However, contradictory conclusions were gathered from another research group looking at adipose tissue samples taken during bariatric surgery. It was concluded that adipose tissue fibrosis was less prevalent in obese patients with diabetes, but adipose hypertrophy was more common in these patients [21]. Additionally, fewer stem cells were found in the adipose tissue samples taken from the diabetic patients [21]. These conflicting results indicate that comorbidities have varying effects on adipose tissue fibrosis and patient demographics should be more closely considered when drawing conclusions. A study comparing healthy, non-obese patients with a predisposition for type 2 diabetes to control patients found that patients with a predisposition had higher degrees of adipocyte hypertrophy, inflammation, and Wnt-signal activation [156]. All these factors contribute to adipose tissue dysfunction. This study further supports that patient demographics should be heavily considered when generalizing adipose tissue fibrosis mechanisms and effects.

In recent years, more diverse patient demographics (race, gender, disease status, etc.) have been used to investigate their correlation to adipose tissue fibrosis. A recent report investigated the link between obesity, insulin resistance, and adipose tissue fibrosis in Chinese Americans [157]. Asian Americans tend to develop type 2 diabetes at lower BMI values compared to White Americans [157,158]. The researchers found that insulin resistance was more strongly correlated to subcutaneous adipose tissue fibrosis than the patient’s body mass index in the Asian American population. Another study examined if there was a difference between patients that had been infected with human immunodeficiency virus (HIV) versus uninfected patients [159]. Patients with HIV related lipoatrophy had a higher degree of fibrosis and adipocyte apoptosis compared to uninfected patients but were unable to conclude if the fibrosis caused insulin resistance [111,159]. Future studies are required to categorize different demographics and their predisposition for adipose tissue fibrosis.

Clinically, adipose tissue fibrosis can be used as a predictor of health-related outcomes, demonstrating the importance of considering it in treatment options. A subcutaneous adipose tissue scoring model was developed to predict weight-loss outcomes after gastric bypass based on the degree of fibrosis in a patient’s adipose tissue [160]. These scores were correlated to the patients’ weight loss after gastric bypass to train a machine learning algorithm, providing semiquantitative, reproducible fibrosis scores. The researchers also determined that the higher the fibrosis score, or the more fibrotic the tissue, the less successful weight-loss was after gastric bypass [160]. Together, these studies reinforce the importance of diagnosing fibrosis in patients to inform better treatment strategies.

## 6. Adipose Tissue’s Role in Treating Other Fibrotic Diseases

Cells sourced from adipose tissue, particularly the SVF, have been used to treat other fibrotic diseases and tissues (Table 2). The SVF consists of all non-adipocyte cell types, including ASCs, endothelial cells, immune cells, and fibroblasts. ASCs are an attractive therapeutic option as they secrete growth factors, cytokines, proteins, and exosomes that promote regeneration [161,162] and have been used to treat systemic sclerosis, as well as dermal, liver, cardiac, renal, muscular, and lung fibrosis [163,164,165,166,167,168,169,170,171,172]. For example, injecting hASCs systemically reduced kidney fibrosis, improved renal functions, and reduced profibrotic gene expression [168,169]. Many of these treatments are effective by inhibiting the inflammatory response through the TGFβ1 signaling pathway [168,171] and reducing collagen deposition [171].

Adipose derived exosomes are another potential therapeutic option. The effect of media conditioned in the presence of ASCs and ASC injections were investigated for treatment of pulmonary fibrosis. Not only did ASCs improve pulmonary fibrosis [173], but secretions in conditioned media also successfully had therapeutic effects [172]. The secretions in the conditioned media included exosomes, which several researchers have used to treat liver fibrosis [174,175]. Exosomes offer a cell free therapy for treating fibrotic diseases. Furthermore, the mechanism is similar, as microRNA181 in the exosomes, reduced TGFβ1 expression, and had in vitro and in vivo therapeutic effects [175].

The significance of these treatments underscores the importance of investigating and modeling adipose tissue fibrosis. These treatments all rely on adipose derived cells; however, adipose tissue fibrosis significantly alters ASCs function and morphology. For these treatments to effectively work a stronger understanding of how adipose tissue fibrosis affects adipose derived stem cells and their secretome at different stages of fibrosis is required.

## 7. Methods for Evaluating Degree of Fibrosis

Detecting and evaluating adipose tissue fibrosis using reproducible, standardized techniques is essential for model development. Information gathered from these techniques in situ can be applied to engineered models to increase their clinical relevance, demonstrating consistency with clinical outcomes. Researchers have used multiple methods to evaluate the presence or degree of fibrosis through staining, imaging, mechanical testing, etc. (Table 3). Each of these techniques offers unique information and has advantages and disadvantages.

## 8. Conclusions

While adipose tissue fibrosis has been investigated for several decades, there is still an incomplete understanding of causes, effects, and mechanisms. Under periods of high caloric consumption, the increase in adipocyte volume and tissue mass results in inflammation, metabolic dysregulation, and adipose tissue fibrosis in some individuals, but not all.

To address many of the open-ended questions surrounding adipose tissue fibrosis, in vitro and in vivo model systems should be carefully chosen to meet the needs of the research question. Three-dimensional in vitro models are an area of adipose tissue fibrosis research that has a high potential for growth. Improvements to this field would drastically enhance our understanding of the disease by creating physiologically relevant models where factors can be carefully manipulated, and treatments can be investigated in human samples. While using a biomaterial scaffold to model adipose tissue fibrosis would offer a high throughput, inexpensive disease model, the tradeoff is a lack of the complexities found in in vivo animal models, including systemic effects and immune responses [81]. Human epidemiological studies vary considerably depending on experimental setup and subject demographics. The effects of race, gender, and comorbidities have just begun to be investigated and appear to have a significant effect on adipose tissue fibrosis [157,158,159]. More in-depth studies and integration of existing datasets [160] are needed to further investigate how patient demographics affect adipose tissue fibrosis.

Currently, there is also a large amount of research in treating other fibrotic diseases using cells and secreted vesicles from adipose tissue. As these treatments rely on cells functioning similarly in all patients the decreased ability to differentiate and exhibit a proinflammatory phenotype in fibrotic tissues is concerning.

Overall, adipose tissue fibrosis is not routinely screened for; however, it can have profound effects on patient outcomes and therapeutic options. With the rate of obesity increasing globally, it is essential that we develop a better understanding of fibrotic causes and mechanisms, as well as create better models to study potential treatments.

## Figures and Tables

**Figure 1 ijms-21-06030-f001:**
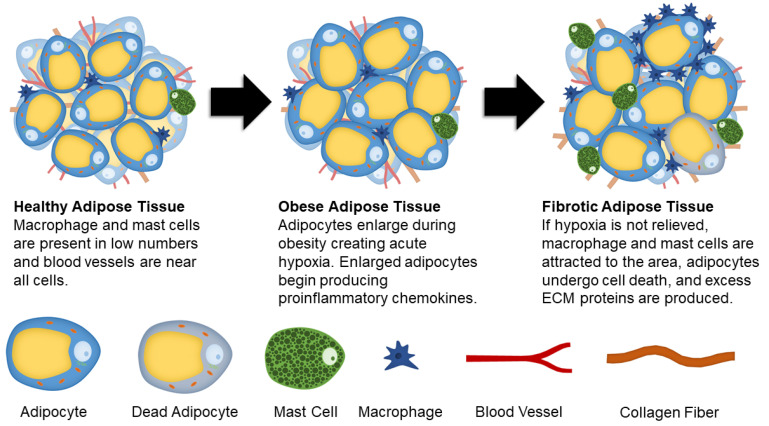
Schematic showing changes to adipose tissue during obesity and fibrosis development.

**Figure 2 ijms-21-06030-f002:**
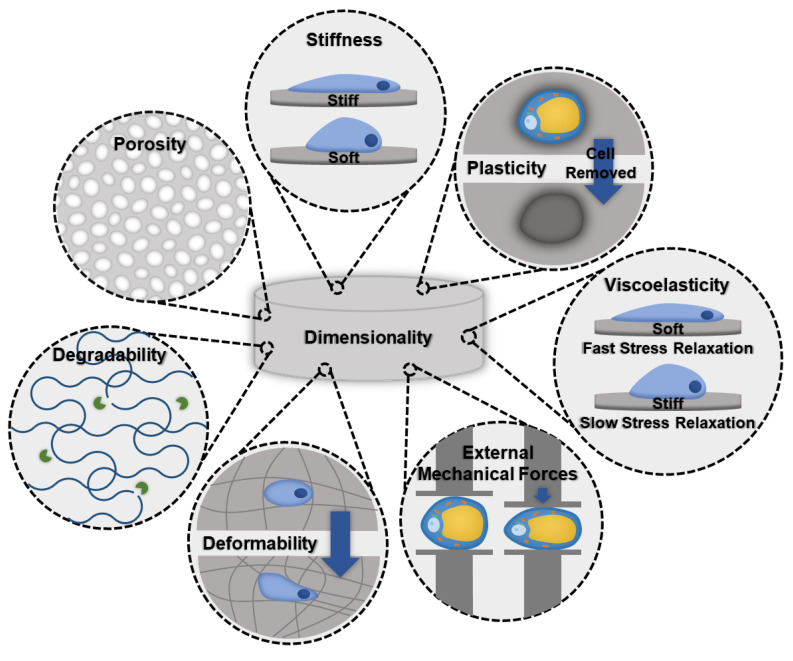
Schematic illustrating important biomaterial properties that should be considered when designing an in vitro fibrotic adipose tissue model.

**Table 1 ijms-21-06030-t001:** Biomaterials that have been used in vitro to culture adipose tissue.

Material	Cell Type	Cell Source	Key Features
Poly (lactic-co-glycolic acid) (PLGA)	ASCs	Rat [83,84] Human [85]	They successfully showed that this PLGA could foster adipose tissue growth and expansion in a short time frame but was not sustainable for long term cultures. At two months post-implantation, there was a dramatic decrease in adipose tissue in the scaffold. This decrease in adipose tissue is likely due to resorption by the environment. One reason for this is lack of vascularization.
	Marrow Stromal Cells (MSCs)	Rat [86]	
Hyaluronic-based Biomaterial (Hyaff-11)	ASCs	Human [87]	Over a month long experiment ASCs were able to mature into adipocytes and showed high cell density [87].
Methacrylated Gelatin and Hyaluronan	ASCs	Human [88]	ASCs were able to remain viable and differentiate into adipocytes in a 4-week time span. This approach was limited by the lack of vascularization [88].
Polycaprolactone (PCL)	Embryonic Stem Cells	Mouse [89]	These scaffolds were composed of nano fibers and had an average pore size of 30 um and porosity of about 88%. The stem cells differentiated into functional adipocytes and used the nano fibers similarly to how they use ECM proteins in vivo.
	ASCs	Human [90]	By comparing random and aligned electrospun PCL fibers to 2D cultures, researchers found that the use of aligned fibers resulted in increased lipid accumulation, decreased proliferation, and closer to physiological glucose uptake in differentiated ASCs.
Bacterial Cellulose Based Biomaterials	MSCs	Mouse [91]	By combining bacterial nanocellulose and alginate, stable porous scaffolds were created. Researchers were able to culture high numbers of adipocytes for 4 weeks.
	ASCS, Microvascular Endothelial Cells	Human [92]	The co-culture of differentiated ASCs and human microvascular endothelial cells was performed with the goal of creating vascularized adipose tissue constructs. Vascular-like structures were seen in co-culture and when culturing endothelial cells only.
Silk Fibroin	ASCs, MSCs	Human [93].	Scaffolds seeded with ASCs or MSCs had higher levels of adipogenesis in vivo compared to collagen and poly (lactic acid) (PLA). In vitro studies showed a comparable level of differentiation of ASCs and MSCs in silk, collagen, and PLA.
	Whole Adipose Tissue (Adipocytes, Stromal Cells, Endothelial Cells)	Human [94,95]	Seeding scaffolds with whole adipose tissue, rather than isolating a specific cell type, creates a more physiologically relevant model. The scaffolds had similar numbers of cells and triglycerides after 3 months of culture compared to after seeding.
Methacrylated Gelatin	MSCs	Human [96]	Three-dimensional printing was used to create microporous methacrylated gelatin scaffolds with varying pore sizes from 230–531 µm. MSCs differentiated in scaffolds regardless of pore size, but there was better spatial distribution and the cells migrated deeper into the scaffolds with the largest pore sizes.
Adipose Tissue ECM	ASCs	Human [97,98,99]	These ECM scaffolds have a number of advantages, such as high pore interconnectivity and mechanical properties optimized for adipose tissue. However, processing can affect the scaffolds biocompatibility and be a complex, lengthy process.

**Table 2 ijms-21-06030-t002:** Synopsis of research that used cells and vesicles sourced from adipose tissue to treat other fibrotic diseases.

	Disease Treated	Model Species	Delivery Method	Outcomes
Adipose Derived Stem Cells	Systemic Sclerosis [163]	Mouse	Intravenous	Decreased skin thickness and collagen content.
	Renal Fibrosis [168,169]	Rat	Intravenous	Improved kidney function and reduced fibrotic tissue.
	Liver Cirrhosis [165]	Mouse	Intravenous	Increased expression of antifibrotic markers
	Cardiac Fibrosis [167,176]	Mouse	Intramyocardial Injection	Improved myocardial function and regeneration.
		Rat		
	Pulmonary Fibrosis [173]	Mouse	Intraperitoneal Injection	Lowered production of profibrotic markers and improved symptoms, such as septal thickening and enlarged alveoli.
	Muscle Fibrosis [171]	Rabbit	Intramuscular	Lowered collagen fiber production and profibrotic markers.
	Dermal Scars [177,178,179]	Human	Subcutaneous Injection	Enhanced tissue regeneration, scar severity and area, and improved the overall cosmetic appearance.
Conditioned Media	Pulmonary Fibrosis [172]	Rat	Intravenous	Found that conditioned media was as effective as ASCs at treating pulmonary fibrosis.
Exosomes	Liver Fibrosis [174,175]	Mouse	Intravenous	Downregulated fibrotic markers and reduced collagen deposits.
			Intrasplenic Injection	
Secretome	Liver Fibrosis [166]	Mouse	Intravenous	Increased expression of antifibrotic, proliferation, and antioxidant activity markers in the liver.

**Table 3 ijms-21-06030-t003:** Techniques used by researchers to evaluate the presence and degree of adipose tissue fibrosis.

Technique	Testing Method	Results
Sampling/Biopsies		Human subcutaneous adipose tissue samples can be gathered from live patients during bariatric surgery or cosmetic procedure (panniculectomy, abdominoplasty, liposuction, etc.) [21,79]. Needle biopsies can also be used while patients are under local anesthesia [79,156].
Staining	Hematoxylin and Eosin	Adipocyte morphology can be determined by staining with hematoxylin and eosin [79].
	Picrosirius Red	Collagen can be detected through histological imaging by staining with Picrosirius red [20,21,160]. The images can be analyzed to determine the collagen and adipocyte area in tissue cross sections.
	Masson’s Trichrome	Using Masson’s trichrome stain allows for collagen, mucus, nuclei, cytoplasm, keratin, muscle fibers, and erythrocytes to be stained [82].
	Pimonidazole hydrochloride	Pimonidazole hydrochloride can be used to stain cells that are in a hypoxic environment [82].
Imaging	Polarized Light Microscopy	Polarized light microscopy can detect different collagen types. Under polarized light and stained with Sirius red, type I collagen fibers will appear orange to red, while type II collagen fibers will appear yellow to green [19,160].
	Confocal Microscopy	Though traditional histological approaches allow important information to be discerned, fully understanding collagen volume and dispersity can only be evaluated using 3D imaging approaches. Confocal microscopy can be used to discern the differences in adipocyte size and collagen amount between healthy and fibrotic tissues [20,81].
	Second Harmonic Generation (SHG) Microscopy	SHG microscopy can be used to image collagen fibers without staining. This allows for 3-dimensional imaging to evaluate collagen dispersion and structure [81,180,181,182]. SHG microscopy has been paired with coherent anti-Stokes Raman scattering (CARS) and 2-photon fluorescence (TPF) to image adipocytes and elastin [183].
	Scanning Electron Microscopy (SEM)	SEM can be used to visualize adipocytes and ECM fibers. Researchers have used SEM to image collagen fibers [28].
	Transmission Electron Microscopy (TEM)	TEM can be used to see the interstitial space, caveolae, vasculature, and adipocytes [28].
Mechanical Testing	Tensile Testing	Tensile testing has been investigated as a method of measuring adipose tissue fibrosis [20]. By securing sections of fresh tissue between clamps the peak force and tensile strength can be determined. Samples with higher degrees of fibrosis will exhibit higher peak forces and tensile strengths. Custom made mechanical testing instruments have also been made [183].
	Rheological Testing	The storage modulus (G’) can be used to measure stiffness on the macroscale [81] which is correlated with increased collagen content.
	Atomic Force Microscopy (AFM)	AFM can be used to quantify stiffness on the microscale [184]. However, AFM only measures the surface stiffness rather than the interior of the sample.
	Shearwave Dispersion Ultrasound Vibrometry (SDUV)	SDUV allows tissue elasticity and viscosity to be measured noninvasively using imaging techniques [185].
	Magnetic Resonance Elastography (MRE)	MRE is a magnetic resonance imaging (MRI) technique. MRE allows for mechanical properties, like stiffness, to be investigated noninvasively using imaging techniques [185].
Gene Expression	PCR, RT-PCR, qPCR, QRTPCR	Specific genes have been linked to adipose tissue fibrosis, such as TGFβ1, αSMA, COL1, and COL6. Biopsied adipose tissue samples can be analyzed through PCR [21,82,154,156].
Assays	Hydroxyproline	Assays can be used to measure the abundance of hydroxyproline, a signature amino acid for fibrillar collagens [82,141].
	Glycerol	The levels of lipolysis can be quantified using a glycerol assay [81]. Lipolysis is thought to be affected by adipose tissue fibrosis with some studies having conflicting results [7,8,9].
Cell Type Frequencies	Flow Cytometry	By staining the cells with specific antibodies flow cytometry can be used to sort cells or count the number of cells in a population. Researchers have used this to quantify the frequency of SVF populations (stem cells, mast cells, and macrophages) [21,58].

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
