# Peer review of "Adipose Tissue Fibrosis: Mechanisms, Models, and Importance"

_ijms, 2020, doi:10.3390/ijms21176030_

Round 1

Reviewer 1 Report

In general, the review is somewhat descriptive with a relative paucity of interpretation of the published data. The sections describing the in vitro models for studying the adipose tissue fibrosis are well written. The sections describing the mechanism of fibrosis and the in vivo model for studying fibrosis are a bit superficial and could have better organization and would benefit from the inclusion of certain citations. Specific comments are as follows:

1. The mechanism of fibrosis section lacks organization. For example, Page 2 line 72: This paragraph talks about collagen I, then different cell types producing collagen and then mentions the role of Collagen VI. The role of different collagens in adipose tissue fibrosis and inflammation should be addressed in a separate subsection while the role of the cell types in producing ECM could be written in the same paragraph. Also, please add a comment on the role of endotrophin along with the role of collagen VI (page 2 line 82).

2. Understanding the role of different cell types in the adipose tissue ECM production has been an active area of research (page 2 line 79). Single-cell RNA seq-based approach in recent studies have identified different cell types and their relative contributions in ECM generation in human and mouse adipose tissues. Following are some of the references:

https://www.nature.com/articles/s42255-019-0152-6?proof=trueInJunhttps://www.ncbi.nlm.nih.gov/pmc/articles/PMC6878222/https://elifesciences.org/articles/49501

Adding a subsection describing the scRNA seq-based approach, whether these studies have identified predominant collagen-producing cell types and how this approach might help us better understand the mechanism of adipose tissue fibrosis will be useful to the readers.

2. Adipose tissue fibrosis is a result of the excess ECM production as well as deranged ECM degradation. Previous studies have addressed the role of TIMPs and MMPs in the development of fibrosis. Please add a subsection addressing the role of collagen degrading enzymes in adipose tissue fibrosis.

3. Please add a comment on the recent paper which identified that adipocytes acquire fibroblast-like gene signature after a high-fat diet: https://www.nature.com/articles/s41598-020-59284-w

Please add one of the key references https://pubmed.ncbi.nlm.nih.gov/28215843/ and comment on the role of adipocyte progenitors in adipose tissue fibrosis.

4. The mechanism section primarily focuses on the role of inflammation and hypoxia as the driving forces for the development of fibrosis. In that context:

a. The role of HIF1α in the regulation of collagen gene expression, and

b. The antagonistic nature of the proinflammatory cytokines and fibrotic gene expression in the setting of obesity needs to be addressed.

6. The in vivo model section should discuss the high-fat diet model of obesity concerning the temporal development and the extent of fibrosis in this mouse model. This section surprisingly does not include the ob/ob mouse model which is one of the most studied models for obesity-associated metabolic derangements and fibrosis.

7. Page 9 line 294: This is not entirely true. Whether any particular gene deletion, such as the deletion of miRNA 155 could be used as a model to study fibrosis is debatable. For example, as discussed in this review, male miRNA155 deleted mice on a high-fat diet develop increased fibrosis. Here, high-fat feeding has been used as a model to generate adipose tissue fibrosis, not the deletion of the miRNA. This is also applicable to other examples in this section, where the role of several genes has been analyzed in the context of HFD-induced obesity-associated fibrosis.

8. Please rewrite ‘in vivo model’ section emphasizing the advantages and the disadvantages of studying HFD-induced obesity, the effect of western diet or cafeteria diet on fibrosis, a general description of monogenic and polygenic models of obesity (genetic models such as the Ob/Ob mice, POMC KO, AgRP KO, s/s mice, etc), a comparative account on diet-induced versus genetic models of obesity and the extent of fibrosis reported in these models. Also, the rodent section should include rat models of obesity (eg., Zucker rats) with a note on the abundance of adipose tissue fibrosis.

8. Please include this reference in the in vitro model section describing matrix stiffness https://pubmed.ncbi.nlm.nih.gov/28926111/

9. Please correct the typographic and sentence organizational errors, for example, page 2 line 45 “investigating the affect adipose tissue fibrosis”, page 3 line 109 “it has been show that”.

Author Response

In general, the review is somewhat descriptive with a relative paucity of interpretation of the published data. The sections describing the in vitro models for studying the adipose tissue fibrosis are well written. The sections describing the mechanism of fibrosis and the in vivo model for studying fibrosis are a bit superficial and could have better organization and would benefit from the inclusion of certain citations. Specific comments are as follows:

Thank you for your excellent comments - they have really improved the quality of this paper.

  1. The mechanism of fibrosis section lacks organization. For example, Page 2 line 72: This paragraph talks about collagen I, then different cell types producing collagen and then mentions the role of Collagen VI. The role of different collagens in adipose tissue fibrosis and inflammation should be addressed in a separate subsection while the role of the cell types in producing ECM could be written in the same paragraph. Also, please add a comment on the role of endotrophin along with the role of collagen VI (page 2 line 82).

Thank you for the suggestion. The information about different collagens was moved to a separate paragraph and more information was added. This really improved the readability of the mechanisms section.

  1. Understanding the role of different cell types in the adipose tissue ECM production has been an active area of research (page 2 line 79). Single-cell RNA seq-based approach in recent studies have identified different cell types and their relative contributions in ECM generation in human and mouse adipose tissues. Following are some of the references:

https://www.nature.com/articles/s42255-019-0152-6?proof=trueInJun, https://www.ncbi.nlm.nih.gov/pmc/articles/PMC6878222/, https://elifesciences.org/articles/49501

Adding a subsection describing the scRNA seq-based approach, whether these studies have identified predominant collagen-producing cell types and how this approach might help us better understand the mechanism of adipose tissue fibrosis will be useful to the readers.

            A section describing scRNA was added. Thank you for the references, they were very useful!

  1. Adipose tissue fibrosis is a result of the excess ECM production as well as deranged ECM degradation. Previous studies have addressed the role of TIMPs and MMPs in the development of fibrosis. Please add a subsection addressing the role of collagen degrading enzymes in adipose tissue fibrosis.

            A section about TIMPs and MMPs was added!

  1. Please add a comment on the recent paper which identified that adipocytes acquire fibroblast-like gene signature after a high-fat diet: https://www.nature.com/articles/s41598-020-59284-w

Please add one of the key references https://pubmed.ncbi.nlm.nih.gov/28215843/ and comment on the role of adipocyte progenitors in adipose tissue fibrosis.

            These papers were added to the review and the discussion on ASCs was expanded.

  1. The mechanism section primarily focuses on the role of inflammation and hypoxia as the driving forces for the development of fibrosis. In that context:
  2. The role of HIF1α in the regulation of collagen gene expression, and
  3. The antagonistic nature of the proinflammatory cytokines and fibrotic gene expression in the setting of obesity needs to be addressed.

            Thank you for the suggestion, this was added.

  1. The in vivo model section should discuss the high-fat diet model of obesity concerning the temporal development and the extent of fibrosis in this mouse model. This section surprisingly does not include the ob/ob mouse model which is one of the most studied models for obesity-associated metabolic derangements and fibrosis.

            Thank you for catching this. We now discuss the use of ob/ob mouse models.

  1. Page 9 line 294: This is not entirely true. Whether any particular gene deletion, such as the deletion of miRNA 155 could be used as a model to study fibrosis is debatable. For example, as discussed in this review, male miRNA155 deleted mice on a high-fat diet develop increased fibrosis. Here, high-fat feeding has been used as a model to generate adipose tissue fibrosis, not the deletion of the miRNA. This is also applicable to other examples in this section, where the role of several genes has been analyzed in the context of HFD-induced obesity-associated fibrosis.

            The study cited in this review used four experimental groups (WT – low fat diet, WT – high fat diet, miRNA155 knockout – low fat diet, and miRNA155 knockout – high fat diet) and determined that their was significantly more fibrosis in the miRNA155 knockout HFD compared to WT HFD. A sentence was added to indicate that miRNA155 deletion and a LFD did not result in fibrosis.

  1. Please rewrite ‘in vivo model’ section emphasizing the advantages and the disadvantages of studying HFD-induced obesity, the effect of western diet or cafeteria diet on fibrosis, a general description of monogenic and polygenic models of obesity (genetic models such as the Ob/Ob mice, POMC KO, AgRP KO, s/s mice, etc), a comparative account on diet-induced versus genetic models of obesity and the extent of fibrosis reported in these models. Also, the rodent section should include rat models of obesity (eg., Zucker rats) with a note on the abundance of adipose tissue fibrosis.

            All wonderful suggestions. We have done accordingly and appreciate the new content in this section.

  1. Please include this reference in the in vitro model section describing matrix stiffness https://pubmed.ncbi.nlm.nih.gov/28926111/

            Thank you for this suggestion. The article was added to the in vitro section. 

  1. Please correct the typographic and sentence organizational errors, for example, page 2 line 45 “investigating the affect adipose tissue fibrosis”, page 3 line 109 “it has been show that”.

               Thank you for reading the article so closely, these have been corrected!

Reviewer 2 Report

Comments to the Authors

This manuscript entitled “Adipose Tissue Fibrosis: Mechanisms, Models, and Importance” is a review article about adipose tissue fibrosis with recent pathophysiological and mechanical information.

Major comment

  1. The authors should have a better understanding of obese adipose tissue. In line 104 and 297, the authors mentioned “crown-like structure” (CLS), but the authors misinterpret the CLS. Adipocyte in CLS is dead or dying one, not obese one. In addition, “macrophage crown-like structures” is incorrect expression. The authors should read the original paper about CLS (J Lipid Res. 49: 1562-8, 2008).
  2. In line 190, there are not αSMA cells, but αSMA-positive cells.

Minor comments

  1. The authors should unify words. Adipose tissue fibrosis vs. adipose fibrosis, Col VI vs. Col6, COL6, wildtype vs. wild-type, knockout vs. knock out, TGF-β1 vs. TGFβ1, and so on.
  2. There are many mistakes in English. The authors should check the manuscript more carefully. In line 88, “Transfoming” is “transforming”, in line 89, “monocyte chemoatactic protein 1” is “monocyte chemoattractant protein 1” (in ref. 30), in line 297, “mouse” is “mice”, in line 330, “fibrosus” is “fibrosis”, and so on. In figure 1, in “Fibrotic Adipose Tissue”, “macrophage” is “macrophages”. In line 285, is “they” correct?
  3. The authors should state the full spell of abbreviations; Lum and Tnc in line 299.
  4. Estrogen receptor α is usually abbreviated ERα.
  5. In line 333, “compared to mice fed a high fat diet” is a little bit difficult to understand, so the reviewer thinks “compared with control mice” is better.
  6. What does the gray shading in the table mean?

Author Response

This manuscript entitled “Adipose Tissue Fibrosis: Mechanisms, Models, and Importance” is a review article about adipose tissue fibrosis with recent pathophysiological and mechanical information.

Major comment

  1. The authors should have a better understanding of obese adipose tissue. In line 104 and 297, the authors mentioned “crown-like structure” (CLS), but the authors misinterpret the CLS. Adipocyte in CLS is dead or dying one, not obese one. In addition, “macrophage crown-like structures” is incorrect expression. The authors should read the original paper about CLS (J Lipid Res. 49: 1562-8, 2008).F

Thank you for listing the original paper. It was read and referenced and the wording on those two phrases was changed.

  1. In line 190, there are not αSMA cells, but αSMA-positive cells.

Thank you for catching that mistake. It has been fixed!

Minor comments

  1. The authors should unify words. Adipose tissue fibrosis vs. adipose fibrosis, Col VI vs. Col6, COL6, wildtype vs. wild-type, knockout vs. knock out, TGF-β1 vs. TGFβ1, and so on.

The wording throughout was changed to increase consistency. Thank you for this comment!

  1. There are many mistakes in English. The authors should check the manuscript more carefully. In line 88, “Transfoming” is “transforming”, in line 89, “monocyte chemoatactic protein 1” is “monocyte chemoattractant protein 1” (in ref. 30), in line 297, “mouse” is “mice”, in line 330, “fibrosus” is “fibrosis”, and so on. In figure 1, in “Fibrotic Adipose Tissue”, “macrophage” is “macrophages”. In line 285, is “they” correct?

Thank you for reading the paper so closely! These mistakes have been fixed.

  1. The authors should state the full spell of abbreviations; Lum and Tnc in line 299.

The full names have been added.

  1. Estrogen receptor α is usually abbreviated ERα.

All cases where αER was written were changed to ERα.

  1. In line 333, “compared to mice fed a high fat diet” is a little bit difficult to understand, so the reviewer thinks “compared with control mice” is better.

This did improve readability! Thank you.

  1. What does the gray shading in the table mean?

The gray shading was intended to help tell the sections apart but during the transferring to the new format the gray was not carried over correctly. This had been fixed.